# Seeking Neural Nuggets: Knowledge Transfer in Large Language Models from a Parametric Perspective

**Ming Zhong[1], Chenxin An[2], Weizhu Chen[3], Jiawei Han[1], Pengcheng He[3]**
[1]University of Illinois Urbana-Champaign, [2]The University of Hong Kong, [3]Microsoft Azure AI
`{mingz5, hanj}@illinois.edu, cxan23@connect.hku.hk`
`wzchen@microsoft.com, Herbert.he@gmail.com`

## Abstract

Large Language Models (LLMs) inherently encode a wealth of knowledge within their parameters through pre-training on extensive corpora. While prior research has delved into operations on these parameters to manipulate the underlying implicit knowledge — encompassing detection, editing, and merging — there remains an ambiguous understanding regarding their transferability across models with varying scales. In this paper, we seek to empirically investigate knowledge transfer from larger to smaller models through a parametric perspective. To achieve this, we employ sensitivity-based techniques to extract and align knowledge-specific parameters between different LLMs. Moreover, the LoRA module is used as the intermediary mechanism for injecting the extracted knowledge into smaller models. Evaluations across four benchmarks validate the efficacy of our proposed method. Our findings highlight the critical factors contributing to the process of parametric knowledge transfer, underscoring the transferability of model parameters across LLMs of different scales. Project website: https://maszhongming.github.io/ParaKnowTransfer.

## 1 Introduction

Driven by the advancements of Large Language Models (LLMs) (Brown et al., 2020; Chowdhery et al., 2022; OpenAI, 2023; Touvron et al., 2023a), a transformative wave has reshaped the landscape in multiple areas of Artificial Intelligence, elevating performance across diverse tasks. From a parametric perspective, the objective of pre-training is to encode substantial amounts of knowledge into model parameters through language modeling on extensive corpora (Peters et al., 2018; Radford et al.; Devlin et al., 2019; Delétang et al., 2023). In a quest to unravel the intricate workings of LLMs, a multitude of research efforts have been directed toward the detailed exploration and nuanced manipulation of this reservoir of implicit knowledge.

Early research efforts sought to detect this parametric knowledge, typically probing the concrete facts by using the "fill-in-the-blank" task under a closed-book setting (Petroni et al., 2019; Jiang et al., 2020; Roberts et al., 2020). Subsequent studies delved into the feasibility of executing operations on model knowledge, including knowledge editing (Cao et al., 2021; Mitchell et al., 2022; Meng et al., 2022), a technique designed to modify targeted knowledge while preserving the integrity of the remaining information, and model merging (Izmailov et al., 2018; Ainsworth et al., 2023; Stoica et al., 2023), a strategy that combines multiple models to enhance robustness or facilitate multitasking abilities. While these investigations exhibited that such parametric knowledge is both *detectable* and *editable* within a single model, the broader question of whether it is *transferable* across different LLMs remains an open and under-explored topic.

Knowledge transfer refers to distilling the expertise of larger teacher models into smaller, more manageable counterparts, thereby democratizing access to cutting-edge machine learning capabilities. As illustrated in Figure 1, online and offline distillation currently stand as the primary paradigms. The former, especially prevalent before the LLM era, capitalizes on teacher models to guide the learning trajectory of student models (Hinton et al., 2015; Sanh et al., 2019; Gou et al., 2021). Yet, as LLMs grow in scale, the inherent demand for the teacher model to undergo fine-tuning or participate in student training becomes increasingly cost-prohibitive. In contrast, offline distillation

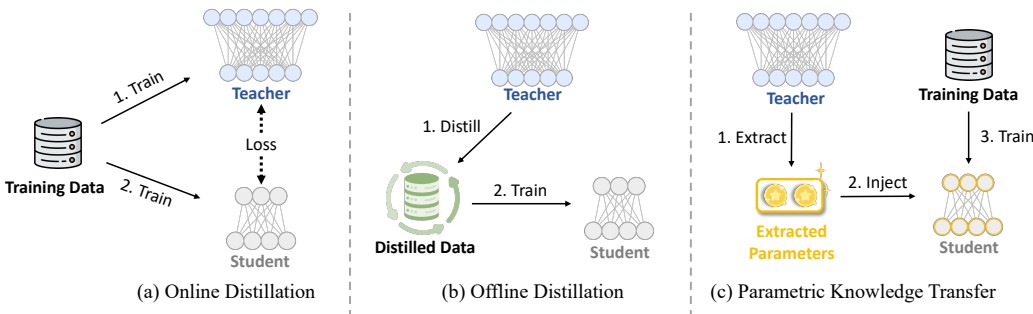

Figure 1: Different paradigms of knowledge transfer from teacher models to student models. (a) Online Distillation: utilizing soft logits from the fine-tuned teacher model to guide the training of the student model; (b) Offline Distillation: generating a distilled dataset that encapsulates the knowledge of the teacher model to fine-tune the student model. (c) Parametric Knowledge Transfer: extracting knowledge-specific parameters from the vanilla teacher model and injecting them into the student model to enhance its training efficacy.

calls upon the teacher model merely to generate answers to open-ended queries, creating a distilled training dataset for students (Honovich et al., 2023; Wang et al., 2023c; Taori et al., 2023). Despite reducing the overhead to thousands of inferences, it completely overlooks the rich knowledge implicitly stored within the teacher's parameters.

In this paper, we empirically investigate knowledge transfer from a distinct parametric perspective, dedicated to selecting static parameters directly from the teacher model and exploring their transferability. Specifically, we introduce a new parametric knowledge transfer paradigm designed to extract task-specific parameters from the teacher model and subsequently inject them into the student model, thereby enhancing performance on downstream tasks. Through decoding on a limited set of seed samples (e.g., 32 samples) with the teacher model, we calculate sensitivity metrics that serve as the basis for knowledge extraction. Given the discrepancies in the number of layers and dimensions across varied LLM scales, we employ sensitivity-based layer mapping and dimensionality reduction techniques to establish alignment between the teacher and student models. Lastly, we leverage LoRA (Hu et al., 2022) as a bridge to inject these extracted parameters into student models, facilitating its fine-tuning on downstream tasks and thus achieving the knowledge transfer process.

Experimentally, we evaluate the parametric knowledge transfer framework across four benchmark categories: reasoning (Cobbe et al., 2021), professional knowledge (Hendrycks et al., 2021), instruction-driven NLP tasks (Wang et al., 2022), and open-ended conversation (Dubois et al., 2023), using various sizes of LLaMA models (Touvron et al., 2023a;b). The results indicate that upon transferring the teacher model's parameters, the student performance demonstrates consistent improvements across all benchmarks, affirming the transferability of parametric knowledge. Furthermore, our detailed analyses illustrate the underlying factors that contribute to effective parametric knowledge transfer, discovering that the teacher scales, initialization strategies, number of seed samples, and the origin and structure of the extracted parameters all play crucial roles.

To summarize, the key contributions of this paper are threefold: (1) From a distinct perspective, we introduce a parametric knowledge transfer paradigm that encompasses stages of sensitivity-based knowledge extraction and LoRA-driven knowledge injection. (2) Through comprehensive evaluations, we provide empirical evidence that implicit model knowledge is indeed *transferable* across varying scales of LLMs. (3) Further enriching our insights into parametric knowledge transfer, we undertake a thorough analysis to pinpoint the pivotal factors that dictate its efficacy.

## 2 RELATED WORK

### 2.1 MANIPULATION OF IMPLICIT MODEL KNOWLEDGE

With the recognition of the vast repository of knowledge embedded in model parameters (Petroni et al., 2019; Jiang et al., 2020; Roberts et al., 2020; Dai et al., 2022), ensuing research has sought to execute diverse operations on these parameters, aiming to manipulate the implicit knowledge.

For instance, knowledge editing endeavors to modify or update specific facts by editing the associated parameters, all the while ensuring the broader knowledge base remains untouched (Cao et al., 2021; Mitchell et al., 2022; Meng et al., 2022; 2023; Yao et al., 2023b). Another avenue, model merging and composition, combines the weights of two or more models into a unified weight set or dynamically activates different modules for greater robustness and multitasking capabilities (Huang et al., 2017; Izmailov et al., 2018; Ainsworth et al., 2023; Stoica et al., 2023; Zhong et al., 2024). Additionally, a strand of studies probes into performing arithmetic operations on the pre-trained weights, thus enabling the model to augment or diminish particular functionalities (Ilharco et al., 2023; Ortiz-Jiménez et al., 2023; Zhang et al., 2023). However, these explorations are limited to operations within individual models or merging models with identical architectures, without investigating if implicit knowledge between different scale models can be manipulated and transferred.

## 2.2 Inheritance of Model Knowledge

Another line of research that aligns more closely with our work concerns the operation of model parameters across scales, specifically the concept of model growth (Yao et al., 2023a; Li et al., 2023). This refers to accelerating the pre-training of LLMs by incrementally growing and expanding the parameters of a smaller model, using them as an initialization for the larger one. The majority of existing work is concentrated on devising function-preserving methods (Chen et al., 2016), ensuring that the initialized larger model replicates the behaviors of the original smaller model (Wei et al., 2016; Gu et al., 2021; Chen et al., 2022; Evci et al., 2022; Shen et al., 2022; Gesmundo & Maile, 2023). Concurrently, several studies adopt data-driven strategies, investigating reverse distillation (Qin et al., 2022) or mapping learned weights from smaller models to their larger counterparts (Wang et al., 2023a). In contrast to this research direction, our emphasis is on the transfer of knowledge from larger teacher to smaller student models, with the aim of exploring not only the efficiency of training, but also the transferability of parametric knowledge across different scenarios.

## 2.3 Transfer of Model Knowledge

Knowledge transfer is a research area dedicated to training a smaller student model to mimic the behavior of a larger pre-trained teacher model to achieve similar performance with fewer parameters (Hinton et al., 2015). Despite progress in improving the online distillation paradigm (Zhang et al., 2018; Lan et al., 2018; Jin et al., 2019; Mirzadeh et al., 2020; Park et al., 2021; Pham et al., 2021; Zhou et al., 2022) and optimizing the efficiency of offline distillation (Honovich et al., 2023; Wang et al., 2023c; Wu et al., 2023; Taori et al., 2023; Peng et al., 2023; Xu et al., 2023), they both completely ignore the implicit knowledge embedded inherently in the teacher model. Concurrently, Xu et al. (2024) propose weight selection for uniformly selecting parameters from a larger teacher model to initialize a smaller variant. In contrast to our work, they concentrate on vision tasks and aim to broadly enhance the capabilities of student models, rather than seeking to affirm the transferability of task-specific implicit knowledge between various models.

## 3 Parametric Knowledge Transfer

In this section, we first outline the task formulation for parametric knowledge transfer. Following this, we delve into a detailed description of our proposed method, as depicted in Figure 2.

### 3.1 Task Formulation

The core objective of parametric knowledge transfer is to enhance a student model by selectively transferring task-specific parametric knowledge from a more knowledgeable teacher model. Given a task $\mathcal{T}$, the transfer process begins with a teacher model $M_T$ endowed with parameters $\mathbf{\Theta}_T$ and a student model $M_S$ characterized by parameters $\mathbf{\Theta}_S$.

The first step in this procedure involves extraction, where task-relevant parameters are identified from the teacher model and resized to a desired scale based on the student model's parameter dimensions. This can be expressed as:

$$\text{Extract}(\mathbf{\Theta}_T; \mathbf{\Theta}_S; \mathcal{T}) = \mathbf{\Theta}_{T_{\text{extract}}}, \tag{1}$$

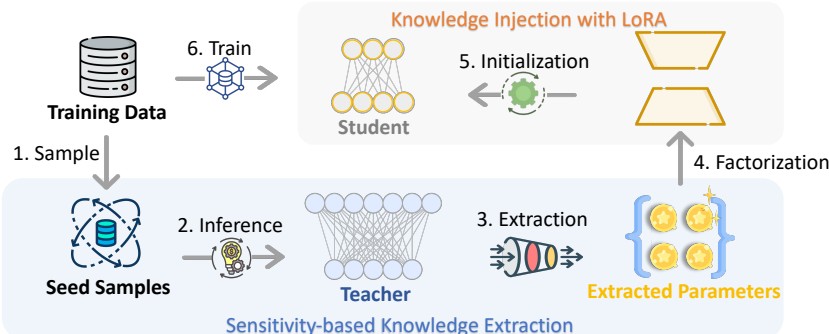

Figure 2: Overview of our parametric knowledge transfer framework. Starting with the teacher model, we compute sensitivity metrics using a set of seed samples, which aids in the extraction of task-specific knowledge. Subsequently, the extracted parameter matrices are factorized to initialize the student model's LoRA module, serving as a bridge for knowledge injection.

with Extract$(\cdot)$ encapsulating the logic for both parameter extraction and downscaling. Following extraction, the next step is the injection of these extracted parameters into the student model:

$$\text{Inject}(\mathbf{\Theta}_S; \mathbf{\Theta}_{T_{\text{extract}}}) = \mathbf{\Theta}'_S, \tag{2}$$

yielding a student model now characterized by the modified parameter set $\mathbf{\Theta}'_S$. Upon completing the knowledge injection, there remains an optional phase wherein the student model fine-tunes the newly incorporated parameters $\mathbf{\Theta}'_S$ with respect to the task $\mathcal{T}$.

### 3.2 KNOWLEDGE EXTRACTION

In implementing our Extract$(\cdot)$ function, we employ parameter sensitivity as the foundational metric to guide the extraction process.

**Sensitivity of the Parameters.** Parameter sensitivity serves as a mechanism to measure the variation in the loss upon setting a particular parameter to zero (Mozer & Smolensky, 1988; Lee et al., 2019; Lubana & Dick, 2021; Liang et al., 2022). When this removal elicits a significant shift in the loss, such a parameter is deemed to be of high sensitivity. For a teacher model $M_T$ with parameters $\mathbf{\Theta}_T = [\theta_1, \ldots, \theta_{N_T}]$, where $N_T$ represents the total number of parameters, the $i$-th parameter can be expressed as $\mathbf{\Theta}_{T_i} = [0, \ldots, \theta_i, \ldots, 0]$. With gradients of the loss relative to $\mathbf{\Theta}_T$ represented as $\nabla_{\mathbf{\Theta}_T}\mathcal{L}$, the sensitivity of the $i$-th parameter for a specific sample $x_j$ from task $\mathcal{T}$ is determined as:

$$S_{i,j} = \left| \mathbf{\Theta}_{T_i}^\top \nabla_{\mathbf{\Theta}_T}\mathcal{L}(x_j) \right|. \tag{3}$$

The rationale behind this sensitivity definition stems from the first-order Taylor expansion of $\mathcal{L}(\cdot)$ relative to $\theta_i$ at $\mathbf{\Theta}_{T_i}$ (Molchanov et al., 2017). In essence, $S_{i,j}$ provides an approximation for how the loss might change in the absence of $\theta_i$:

$$\mathbf{\Theta}_{T_i}^\top \nabla_{\mathbf{\Theta}_T}\mathcal{L}(x_j) \approx \mathcal{L}(\mathbf{\Theta}_T) - \mathcal{L}(\mathbf{\Theta}_T - \mathbf{\Theta}_{T_i}). \tag{4}$$

To ascertain $S_i$ for parameter $i$ pertaining to task $\mathcal{T}$, we randomly sample $k$ instances as seed samples for an efficient and representative estimate. Thus, the final formulation $S_i$ for task $\mathcal{T}$ integrates the cumulative sensitivity over the sampled instances, calculated as $S_i = \sum_{j=1}^{k} S_{i,j}$.

**Layer Selection and Dimensionality Reduction.** Given that models of varying scales often differ in both the number of layers and their dimensions, we adopt a method of layer selection and dimensionality reduction based on sensitivity scores. Our first step is to assess the layers of the teacher model, $M_T$, with respect to their relevance to task $\mathcal{T}$. Let $L_T$ and $L_S$ represent the total number of layers in the teacher and student models, respectively, with $L_S \leq L_T$. For each layer $l$ in $M_T$, we calculate a layer-specific sensitivity score, $S_{T_l}$, by aggregating the sensitivity scores of all parameters within that layer, represented as: $S_{T_l} = \sum_{\theta_i \in \mathbf{\Theta}_{T_l}} S_i$. Having computed the sensitivity scores

for layer $l$ in $M_T$, we proceed to arrange them in descending order and select the top $L_S$ layers with the highest scores. To preserve the inherent structure of the teacher model, the chosen layers are subsequently mapped to the student model maintaining their original sequential order.

Upon alignment of the layers, the parameter dimensions of each layer in the teacher model typically continue to surpass those of the student model. During this phase of the transfer process, our focus is primarily on all the two-dimensional matrices in the teacher model, which are denoted as $\mathbf{W_T}$. We initially identify submatrices within each $\boldsymbol{W}_{T_i} \in \mathbf{W_T}$ that match the student model's matrix dimensions, $\mathbb{R}^{n_S \times m_S}$ ($n_S \leq n_T, m_S \leq m_T$). The extraction of these submatrices can be conducted through different methods: pass-by-neuron, by selecting rows and columns, or by direct extraction of the submatrix. Our comparative analysis in Section 4.3 indicates that maintaining the original structural integrity of the teacher model's parameters is most effective. Consequently, we compute the sensitivity scores for all submatrices with dimensions $\mathbb{R}^{n_S \times m_S}$. The submatrix $\boldsymbol{W}_{T_i,\text{extract}}$ is then selected as the one with the highest cumulative sensitivity score among these. Formally, this objective is expressed as:

$$\boldsymbol{W}_{T_i,\text{extract}} = \arg \max_{\boldsymbol{W}' \subseteq \boldsymbol{W}_{T_i}} \sum_{\theta_i \in \boldsymbol{W}'} S_i. \tag{5}$$

By aggregating $\boldsymbol{W}_{T_i,\text{extract}}$ across all layers, we derive the extracted parameters $\boldsymbol{\Theta}_{T_{\text{extract}}}$ from $M_T$.

## 3.3 KNOWLEDGE INJECTION

To keep the architecture and the number of parameters of the student model unchanged during the knowledge transfer process, we employ the LoRA approach to instantiate the Inject($\cdot$) function.

**LoRA Module.** LoRA (Hu et al., 2022), which stands for Low-Rank Adaptation, is a method designed to optimize parameter efficiency by freezing the pre-trained model weights and inserting trainable rank decomposition matrices into the deep neural network. The guiding intuition is that pre-trained language models possess low "intrinsic dimensions" (Aghajanyan et al., 2021). This means that even when projected to a smaller subspace, these models can still exhibit efficient learning. Consequently, it can be hypothesized that weight updates during adaptation also exhibit low "intrinsic ranks". For a given pre-trained weight matrix $\boldsymbol{W}_i \in \mathbb{R}^{n \times m}$, it can be updated as:

$$\boldsymbol{W}_i^* = \boldsymbol{W}_i + \Delta \boldsymbol{W} = \boldsymbol{W}_i + \boldsymbol{B}_i \boldsymbol{A}_i, \quad \boldsymbol{B}_i \in \mathbb{R}^{n \times r}, \quad \boldsymbol{A}_i \in \mathbb{R}^{r \times m}, \tag{6}$$

where $r$ represents the low rank with $r \ll \min(n, m)$. The matrix $\boldsymbol{W}_i$ remains constant during this operation, implying that only $\boldsymbol{B}_i$ and $\boldsymbol{A}_i$ are updated in the training phase. To ensure that training commences from the original pre-trained weights, either $\boldsymbol{B}_i$ or $\boldsymbol{A}_i$ is initialized with zeros.

**Knowledge Injection with LoRA.** The main goal of this step is to integrate knowledge from the teacher model by incorporating extracted parameters into the student's LoRA module. Initially, SVD is adopted to factorize each matrix $\boldsymbol{W}_{T_i,\text{extract}}$ in $\boldsymbol{\Theta}_{T_{\text{extract}}}$ into three constituent matrices as:

$$\boldsymbol{W}_{T_i,\text{extract}} = \boldsymbol{U}_i \boldsymbol{\Sigma}_i \boldsymbol{V}_i^\top. \tag{7}$$

Here, $\boldsymbol{U}_i$ and $\boldsymbol{V}_i^\top$ are orthogonal matrices containing left and right singular vectors, respectively, while $\boldsymbol{\Sigma}_i$ is a diagonal matrix that hosts the singular values in descending order. To capture the first $r$ ranks, we then formulate:

$$\boldsymbol{W}_{T_i,\text{extract},r} = \boldsymbol{U}_i[:,:r]\boldsymbol{\Sigma}_i[:r,:r]\boldsymbol{V}_i^\top[:r,:]. \tag{8}$$

The symbols $\boldsymbol{U}_i[:,:r]$ and $\boldsymbol{V}_i^\top[:r,:]$ represent the initial $r$ columns of $\boldsymbol{U}_i$ and $\boldsymbol{V}_i^\top$, respectively, while $\boldsymbol{\Sigma}_i[:r,:r]$ captures the top $r$ singular values. For the student model's corresponding matrix $\boldsymbol{W}_i$, we can adopt the training strategy from the LoRA paper:

$$\boldsymbol{W}_i^* = \boldsymbol{W}_i + \boldsymbol{B}_i \boldsymbol{A}_i, \tag{9}$$

where $\boldsymbol{B}_i$ is initialized as $\boldsymbol{U}_i[:,:r]\boldsymbol{\Sigma}_i[:r,:r]$, and $\boldsymbol{A}_i$ with $\boldsymbol{V}_i^{\top}[:r,:]$. This approach, however, alters the starting point from the pre-trained weights of the student model, potentially impacting downstream task performance, as discussed in Section 4.3. Consequently, we propose an alternative initialization strategy for the student model:

$$\boldsymbol{W}_i^* = \boldsymbol{W}_i - \boldsymbol{W}_{T_i,\text{extract},r} + \boldsymbol{B}_i\boldsymbol{A}_i, \tag{10}$$

During the training process, we maintain the matrices $\boldsymbol{W}_i$ and $\boldsymbol{W}_{T_i,\text{extract},r}$ as constants, with updates only being applied to the parameters in $\boldsymbol{B}_i\boldsymbol{A}_i$. Given that $\boldsymbol{W}_{T_i,\text{extract},r}$ and $\boldsymbol{B}_i\boldsymbol{A}_i$ are initially equivalent, this approach guarantees that training commences from the pre-trained weights. The inclusion of the LoRA module is designed to efficiently utilize the most salient features of the extracted knowledge from the teacher model.

## 4 EXPERIMENTS

### 4.1 EXPERIMENTAL SETUP

**Facets of Evaluation.** To validate the efficacy of our proposed framework, we conduct evaluations across four distinct benchmark categories:

(1) Reasoning: Reasoning stands as a foundational capability for models, particularly when tackling intricate tasks. We leverage the Grade School Math dataset (GSM) (Cobbe et al., 2021) to assess the reasoning proficiency of models. The evaluation format requires models, given a math problem, to produce the chain-of-thought process (Wei et al., 2022) and the final numerical answer.

(2) Professional Knowledge: For language models to effectively cater to users' informational needs, possessing a robust repository of professional knowledge is crucial. We measure this knowledge reservoir using the Massive Multitask Language Understanding dataset (MMLU) (Hendrycks et al., 2021). This dataset encompasses questions about 57 subjects, spanning a spectrum of difficulty levels from elementary to advanced professional tiers, all formatted as multiple-choice questions.

(3) Instruction-driven NLP Tasks: This set of tasks evaluates a model's capability in adhering to instructions. Typically, the language model receives both a task definition and an input text, and it must perform the specified classification or generation tasks as directed. Our chosen benchmark for this category is the Super Natural Instructions (Super NI) (Wang et al., 2022), a rich dataset comprising 1,616 varied NLP tasks alongside their expert-written instructions.

(4) Open-ended Conversation: This represents the primary interface through which models interact with users in real-world scenarios. To evaluate such instructability, we employ AlpacaFarm (Dubois et al., 2023), which contains 805 instructions including subsets from various evaluations like Self-Instruct (Wang et al., 2023c), Open Assistant (Köpf et al., 2023), Anthropic (Bai et al., 2022), Vicuna (Chiang et al., 2023), and Koala (Geng et al., 2023). GPT4-32K serves as the evaluator, determining the win rate of the testing model against the outputs generated by Davinci-003.

Throughout all evaluations, we adhere to established metrics and prompts, utilizing the evaluation scripts sourced from Wang et al. (2023b).

**Implementation Details.** For all our experiments, the larger-scale LLaMA model (Touvron et al., 2023a;b) serves as the teacher, and its smaller-scale counterpart acts as the student. For the fine-tuning of the student model, we draw a random subset of 1,000 instances from the respective training datasets of each benchmark. In the case of AlpacaFarm, due to the absence of a training set, we utilize LIMA data (Zhou et al., 2023) as a substitute, which is composed of 1,000 carefully curated open-ended conversations. For each experiment, 32 seed samples are randomly selected from the corresponding training sets. The student model is trained for 3 epochs with a batch size of 64 and a learning rate of 3e-4. Regarding LoRA, we set the rank as 16, and insert LoRA module into the embedding layer, FFN, and self-attention layer in the Transformer architecture (Vaswani et al., 2017). Notably, all results presented in this paper are mean values derived from three separate runs, with each run using a new random set of seed samples.

Table 1: Results for parametric knowledge transfer. "7B-LoRA + 13B Param." represents that we extract parameters from the 13B teacher model and transfer them to the 7B student model.

| Models | GSM | | MMLU | | Super NI | | AlpacaFarm | Average |
|---|---|---|---|---|---|---|---|---|
| | 0-shot | 8-shot | 0-shot | 5-shot | EM | R-L | Win Rate% | - |
| *LLaMA-1* | | | | | | | | |
| Vanilla 7B | 4.70 | 10.77 | 32.10 | 35.30 | 0.67 | 5.55 | - | - |
| 7B-LoRA | 17.26 | 16.93 | 43.43 | 38.90 | 22.91 | 40.49 | 9.07 | 27.00 |
| + 13B Param. | **18.73** | **18.85** | 44.03 | 39.77 | 24.51 | 42.37 | 9.28 | 28.22 |
| + 30B Param. | 18.63 | 18.52 | **45.20** | **40.60** | **25.01** | **43.08** | **9.40** | **28.63** |
| Vanilla 13B | 4.93 | 17.44 | 43.50 | 46.80 | 2.18 | 7.78 | - | - |
| 13B-LoRA | 26.18 | 23.78 | 50.43 | 50.03 | 27.34 | 45.53 | 13.91 | 33.89 |
| + 30B Param. | **27.85** | **27.70** | **51.30** | **51.03** | **27.51** | **46.09** | **17.27** | **35.54** |
| *LLaMA-2* | | | | | | | | |
| Vanilla 7B | 3.34 | 15.54 | 41.70 | 45.80 | 0.00 | 4.68 | - | - |
| Vanilla 13B | 6.52 | 27.82 | 52.10 | 55.20 | 0.00 | 4.84 | - | - |
| 7B-LoRA | 23.38 | 21.05 | 47.77 | **47.07** | 24.93 | 41.25 | 20.50 | 32.28 |
| + 13B Param. | **25.30** | **26.31** | **49.37** | 46.53 | **26.16** | **42.98** | **24.64** | **34.47** |

## 4.2 EXPERIMENTAL RESULTS

**Results for Parametric Knowledge Transfer.** Our initial experiments focus on four distinct teacher-student model pairings: LLaMA-1 (13B $\Rightarrow$ 7B, 30B $\Rightarrow$ 7B, 30B $\Rightarrow$ 13B) and LLaMA-2 (13B $\Rightarrow$ 7B). The outcomes are systematically presented in Table 1. Remarkably, in contrast to the direct fine-tuning approach of LoRA, the student models augmented with parametric knowledge from their respective teacher models exhibit substantial improvements across all four benchmark categories. For instance, the LLaMA-1 30B $\Rightarrow$ 7B pairing yields an average performance boost of 6.04% (from 27.00 to 28.63). In a similar vein, the LLaMA-2 13B $\Rightarrow$ 7B configuration brings an enhancement of 6.78% (from 32.28 to 34.47).

Another observation emerges when examining the effects of scaling up the teacher model, specifically transitioning from 13B to 30B. The performance of the student model, LLaMA-1 7B, generally sees an improvement, despite a slight decrement in the GSM benchmark. Beyond the evident performance gains, the overhead introduced by parametric knowledge transfer remains minimal. The only extra commitment involves the teacher model executing inference on a set of 32 seed samples, without any direct participation in the training. Considering both performance and efficiency, parametric knowledge transfer stands out as a practical technique, even as disparities in parameter counts and architectural variances between teacher and student models expand.

Table 2: Transfer experiments with different task-specific extracted parameters. The leftmost column indicates the dataset on which the knowledge extraction is based. The teacher model and student model are LLaMA-2 13B and 7B, respectively.

| Models | GSM | | MMLU | | Super NI | | AlpacaFarm | Average |
|---|---|---|---|---|---|---|---|---|
| | 0-shot | 8-shot | 0-shot | 5-shot | EM | R-L | Win Rate% | - |
| Vanilla 7B | 3.34 | 15.54 | 41.70 | 45.80 | 0.00 | 4.68 | - | - |
| 7B-LoRA | 23.38 | 21.05 | 47.77 | 47.07 | 24.93 | 41.25 | 20.50 | 32.28 |
| GSM | **25.30** | **26.31** | 48.40 | 45.97 | 24.45 | 42.11 | 23.68 | 33.75 |
| MMLU | 24.11 | 25.47 | **49.37** | 46.53 | 25.55 | 42.55 | 24.01 | 33.94 |
| Super NI | 23.78 | 24.11 | 48.60 | 46.70 | **26.16** | **42.98** | 24.31 | 33.81 |
| LIMA | 24.08 | 25.60 | 49.03 | **47.23** | 25.63 | 42.83 | **24.64** | **34.15** |

**Transfer Experiments with Task-specific Extracted Parameters.** While our results indicate that transferring extracted knowledge from the teacher model positively influences student model performance, the nature of this improvement—whether it is rooted in generalized knowledge or task-specific expertise—warrants deeper exploration. To disentangle this, we conduct experiments

wherein extracted parameters, each tailored to a specific task, are integrated into the student model, which is subsequently fine-tuned across all datasets.

Table 2 offers insights into a prevalent trend: when parameters are extracted from a concrete task, the performance is most significantly amplified for that same task. This is particularly evident in the GSM benchmark. Models equipped with GSM-oriented extracted parameters notably exceed their counterparts—achieving at least a 1.2 increase in 0-shot accuracy—compared to models incorporated with parameters based on alternative datasets. This is likely due to the unique and intricate challenges associated with mathematical reasoning. Additionally, parameters sourced from the LIMA dataset demonstrate remarkable generalizability, presumably owing to their grounding in open-ended dialogues that cover a spectrum of domains and tasks. Overall, these observations highlight the capability of our sensitivity-driven techniques to efficiently target certain types of knowledge, rather than just extracting generalized knowledge.

## 4.3 ANALYSIS: KEY FACTORS FOR PARAMETRIC KNOWLEDGE TRANSFER

We further analyze the key factors for the process of parametric knowledge transfer as follows.

**Initialization Strategies.** Our analysis begins with a comparison of two initialization strategies: the approach described in the LoRA paper (Equation 9) and our proposed method (Equation 10). We employ the LLaMA-1 7B as the student model and explore 5 methods to initialize its LoRA module. These include Gaussian initialization for both $B$ and $A$ matrices, random extraction of sub-matrices from the 13B and 30B models, and sensitivity score-based extraction of sub-matrices from both the 13B and 30B models.

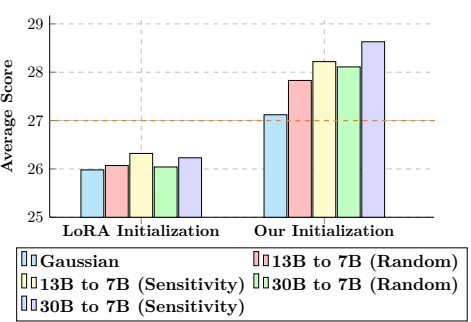

Figure 3: Comparison of different initialization strategies. The y-axis represents the average score over four datasets. "13B to 7B (Sensitivity)" refers to initializing the LoRA module in the 7B model with submatrices from the 13B model based on sensitivity score. The orange dotted line denotes the result of fine-tuning 7B-LoRA without knowledge transfer.

We present our findings in Figure 3. Initializing as per the LoRA paper—but without zeroing out $BA$—leads to a noticeable drop in performance. Recognizing the imperative of leveraging the original model's weights as a starting point for fine-tuning the LoRA, our initialization strategy in this paper is rooted in Equation 10; hence, we keep both $W$ and $W_{\text{extract}}$ fixed and solely fine-tune $BA$. Moreover, our sensitivity-based method consistently outperforms both Gaussian initialization and random parameter extraction from teacher models across varying scales.

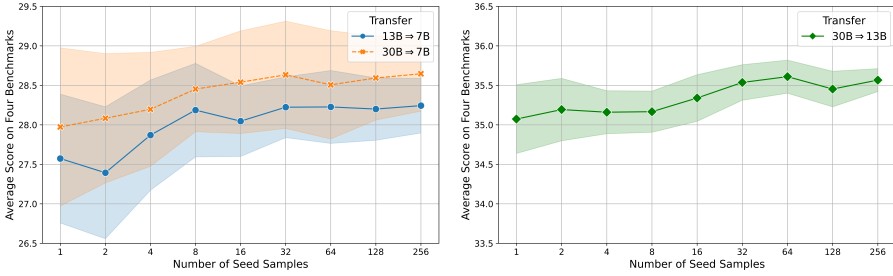

Figure 4: Analysis of how the quantity of seed samples affects student performance.

**Number of Seed Samples.** The quantity of seed samples plays a crucial role in determining both the reliability and efficiency of computing sensitivity scores from the teacher model. To delve deeper

into its impact, we study how varying numbers of seed samples influence the performance of the student model. As evidenced in Figure 4, an augmentation in seed samples consistently mitigates variance, whilst the enhancement in performance remains relatively slight. The results demonstrate a tendency to stabilize after the application of 32 seed samples, prompting us to establish this as a hyperparameter in this paper. A further insight is the marked reduction in variance as the student's scale is escalated (transitioning from 30B ⇒ 7B to 30B ⇒ 13B), or as the disparity between the teacher and student models is diminished (transitioning from 30B ⇒ 7B to 13B ⇒ 7B).

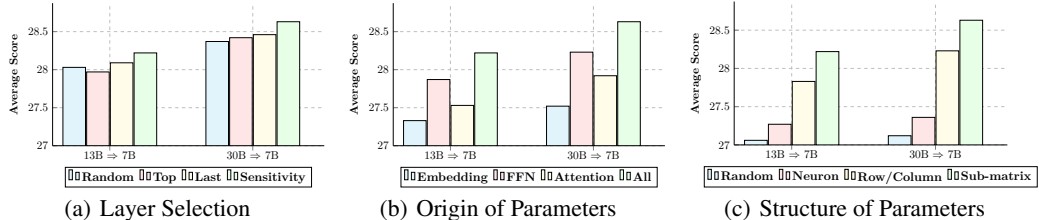

(a) Layer Selection          (b) Origin of Parameters          (c) Structure of Parameters

Figure 5: Analysis of various aspects of extracted parameters from teacher models. The y-axis begins with the result of direct fine-tuning students without knowledge injection.

**Layer Selection Methods.**    Owing to the discrepancy in the number of layers between teacher and student models, the selection methodology for these layers potentially influences the final results. We evaluate four strategies: random layer selection, extracting the top or last layers, and a selection based on our sensitivity-centric technique. In the experiments, we consistently map teacher layers to student layers in their inherent sequential order. As Figure 5(a) illustrates, while the layer selection modestly affects student performance, our sensitivity-driven approach excels over the other strategies across both teacher-student model pairings.

**Origin of Extracted Parameters.**    The complex architecture of the Transformer raises inquiries about the most effective module for knowledge transfer. Our explorations involve the Embedding layer, the Feed-Forward Network (FFN), and the Self-Attention layer of the teacher model. As depicted in Figure 5(b), the embedding layer experiences inferior transfer effectiveness, likely due to its lesser parameter quantity. In contrast, the FFN showcases advanced transfer capabilities, intimating that it houses a significant share of the teacher's knowledge. Optimal results are obtained when transferring knowledge from all available modules.

**Structure of Extracted Parameters.**    The necessity to reduce the parameter matrix's size for knowledge transfer prompts questions regarding optimal population strategies for this matrix. We undertake a comparison across four methods: random single-weight selection from the teacher model, and parameter extraction based on the highest sensitivity at the single weight, row and column, and submatrix levels. Figure 5(c) shows that maintaining the teacher model's parameter structure significantly benefits student model performance. More precisely, transferring isolated single weights—either randomly or based on sensitivity—yields results comparable to those without knowledge transfer, highlighting the ineffectiveness of such transfers. Preserving the coherence of rows or columns provides a notable improvement, and the preservation of the submatrix structure further augments the performance gains derived from parametric knowledge transfer. This observation underpins our proposed knowledge extraction approach as outlined in Equation 5.

## 5    CONCLUSION

In this paper, we delve into the feasibility of transferring parametric knowledge between LLMs of varying scales, and present a new paradigm, exploring knowledge transfer from a distinct parametric perspective. Through our two-stage framework encompassing knowledge extraction and injection, we perform extensive experiments across four diverse benchmarks, affirming the inherent transferability of model parameters. Furthermore, by meticulously analyzing the key elements influencing parametric knowledge transfer, we aim to shed light on future research in this domain.

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

## A APPENDIX

### A.1 VISUALIZATION FOR PARAMETRIC KNOWLEDGE

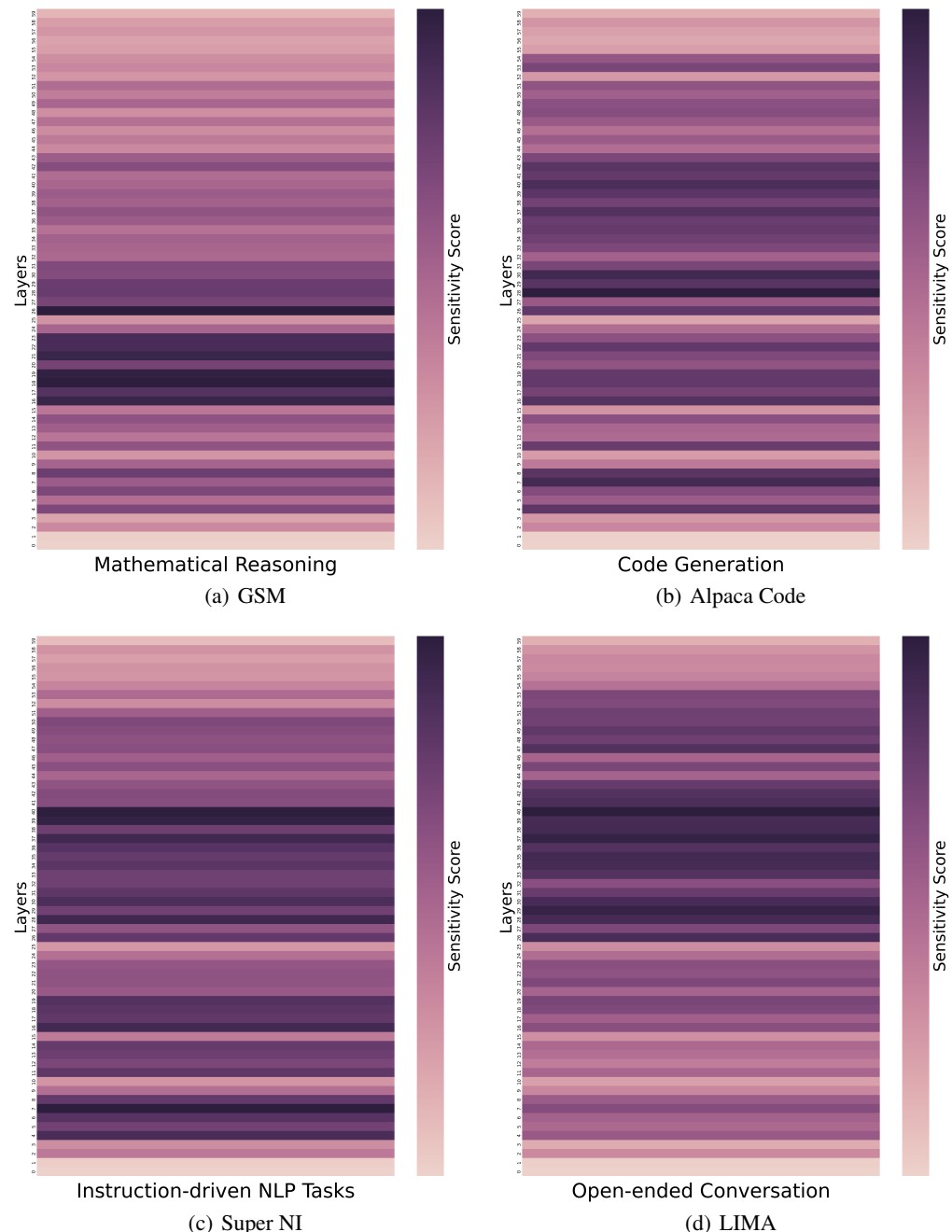

Figure 6: Visualization of parametric knowledge across different layers for four distinct task categories. Darker shades represent higher sensitivity scores for each layer.

In our exploration, we also attempt to visualize the parametric knowledge intrinsic to different task categories. MMLU is omitted from the set of tasks, given its encompassing knowledge from multiple domains, and we introduce code generation (Chaudhary, 2023) as an additional task for analysis. LLaMA-1 30B serves as the teacher model, and we base our findings on 32 randomly selected seed samples, illustrating the sensitivity scores layer by layer. During the visualization process, we sub-

ject each parameter matrix to min-max normalization, ensuring that sensitivity scores fall within the [0, 1] range. The insights from Figure 6 reveal that the distribution of parametric knowledge across layers varies considerably among tasks. For instance, mathematical reasoning predominantly engages the bottom layer, instruction-driven NLP tasks concentrate on the bottom and middle layers, open-ended conversations are more centered around the middle and upper layers, while code generation appears to draw from all layers. This further emphasizes the efficacy of our sensitivity-based knowledge extraction method in pinpointing task-specific parametric knowledge, thereby aiding the subsequent transfer processes between diverse models.

## A.2 EXTENDED EXPERIMENT ON LLAMA-2 70B TO 7B

Table 3: Results for parametric knowledge transfer on LLaMA-2, with additional 70B to 7B transfer results included.

| Models | GSM (0-shot) | MMLU (0-shot) | Super NI (R-L) | AlpacaFarm | Average |
|---|---|---|---|---|---|
| Vanilla 7B | 3.34 | 41.70 | 4.68 | - | - |
| Vanilla 13B | 6.52 | 52.10 | 4.84 | - | - |
| 7B-LoRA | 23.38 | 47.77 | 41.25 | 20.50 | 33.23 |
| + 13B Param. | 25.30 | 49.37 | 42.98 | **24.64** | **35.57** |
| + 70B Param. | **26.16** | **49.60** | **43.65** | 24.27 | **35.92** |

As shown in Table 3, we additionally exhibit the LLaMA-2 70B to 7B experiment for reference. Due to the differences in attention architectures between LLaMA-2 70B and 7B models (with the 70B employing grouped-query attention and the 7B using multi-head attention), we restrict the parametric knowledge transfer from 70B to 7B to the FFN and embedding layers, which account for 0.36% trainable parameters. In contrast, in the transfer from the 13B to 7B model, we also include the attention module, increasing the percentage of trainable parameters to 0.61%. Nevertheless, the transfer from the 70B to the 7B model demonstrates greater performance gains than transferring from 13B to 7B. This implies that our parametric knowledge transfer approach becomes increasingly effective as the teacher model scales up, even in the presence of architectural differences beyond the number of layers and dimensions.

## A.3 ANALYSIS OF THE NUMBER OF PARAMETERS

Table 4: Analysis of the effect of the number of parameters on the performance of parametric knowledge transfer.

| Transfer Module | LoRA r | Params. | 13B to 7B | 30B to 7B |
|---|---|---|---|---|
| Embedding | 128 | 0.137% | 27.33 | 27.52 |
| Attention | 16 | 0.248% | 27.53 | 27.92 |
| FFN | 16 | 0.343% | 27.87 | 28.23 |
| All Modules | 4 | 0.152% | 27.93 | 28.37 |
| All Modules | 8 | 0.304% | 28.17 | 28.48 |
| All Modules | 16 | 0.608% | 28.22 | **28.63** |
| All Modules | 32 | 1.216% | 28.19 | 28.54 |
| All Modules | 64 | 2.432% | **28.27** | 28.60 |

Our analysis of the origin of parameters (see Figure 5(b)) involves a discussion of parameter amounts. For example, transferring a combination of the embedding layer, FFN, and attention modules, which constitute 0.608% of the model's parameters, yields the best results. In contrast, transferring a single module, like the FFN alone which accounts for 0.343% trainable parameters, leads to relatively poor performance.

To further investigate the effect of the number of parameters, we extend the experiments at LLaMA-1 13B to 7B and 30B to 7B by introducing comparisons with different LoRA r (intrinsic rank). The results of the average score on the four datasets are listed in Table 4. For the transfer involving

only the embedding layer, we set LoRA $r$ at 128, ensuring that the number of trainable parameters remains comparable. We observe that increasing LoRA $r$ beyond 16 does not significantly enhance the results when transferring all modules. Consequently, we maintain LoRA $r$ at 16 for the experiments presented in this paper. Our analysis indicates that the origin of the parameters has a more pronounced impact on the effectiveness of parametric knowledge transfer than the number of parameters transferred.

### A.4 COMPARISON WITH DISTILLATION METHODS

The parametric knowledge transfer paradigm in this paper is fundamentally distinct from traditional distillation methods, characterized by the following differences:

- **Purpose and Focus**: Rather than proposing better distillation methods, our study is focused on exploring the transferability of implicit knowledge embedded in static parameters. While prior research has explored the detectability and editability of parametric knowledge, its transferability remains less explored. Our experiments provide empirical evidence in the knowledge transfer scenario, where student models show improved performance after receiving task-specific knowledge from the teacher model, as shown in Tables 1 and 2.
- **Process and Efficiency**: Our approach differs from standard knowledge distillation, which typically requires fine-tuning or the direct involvement of the teacher model in student model training — a computationally intensive process. In contrast, our parametric knowledge transfer involves extracting task-specific parameters from the vanilla teacher model and integrating them into the student model. This method, requiring only 32 inferences from the teacher model, offers a significant efficiency advantage, especially in the context of LLMs.

Despite these differences, we provide the results of the distillation methods as a reference in the knowledge transfer scenario. Table 1 contains the results for LLaMA-1 13B to 7B and 30B to 7B, where KD refers to vanilla knowledge distillation (Hinton et al., 2015) and SeqKD is sequence-level knowledge distillation (Kim & Rush, 2016).

Table 5: Results for parametric knowledge transfer on LLaMA-1, additionally including results from knowledge distillation methods.

| Models | GSM (0-shot) | MMLU (0-shot) | Super NI (R-L) | AlpacaFarm | Average |
|---|---|---|---|---|---|
| Vanilla 7B | 4.70 | 32.10 | 5.55 | - | - |
| Vanilla 13B | 4.93 | 43.50 | 7.78 | - | - |
| 7B-LoRA | 17.26 | 43.43 | 40.49 | 9.07 | 27.56 |
| 13B to 7B (KD) | 17.69 | 43.57 | 42.08 | 9.32 | 28.17 |
| 13B to 7B (SeqKD) | 17.86 | 43.33 | 41.91 | 9.36 | 28.12 |
| 13B to7B (Ours) | **18.73** | 44.03 | 42.37 | 9.28 | 28.60 |
| 30B to 7B (KD) | 17.81 | 44.10 | 41.96 | 9.48 | 28.34 |
| 30B to 7B (SeqKD) | 17.99 | 43.97 | 42.40 | **9.61** | 28.49 |
| 30B to 7B (Ours) | 18.63 | **45.20** | **43.08** | 9.40 | **29.08** |

All models are fine-tuned with LoRA, using identical training data and hyperparameters. In this paper, we attempt to explore the evidence that knowledge in static parameters can be transferred between different LLMs, and knowledge transfer is the scenario in which we find and provide empirical evidence. Our focus is not on proposing to find better methods in this scenario.

### A.5 TRADE-OFF DISCUSSION BETWEEN PERFORMANCE AND RUNNING COST

Considering that users may have varying computational resources in practical application scenarios, we discuss the trade-offs between performance and running costs as follows:

Experimental details:

- We conduct comparisons for two knowledge transfers: LLaMA-1 13B to 7B and 30B to 7B.

Table 6: Experimental results for the trade-off discussion between the performance and the running cost.

| Transfer | Structure of Ext. Para. | Methods for Ext. Para. | Score | Inference Time | Memory |
|---|---|---|---|---|---|
| 13B to 7B | Submatrix | Random | 27.83 | 39 min | 205 G |
| 13B to 7B | Submatrix | Sensitivity | 28.22 | 39 min | 205 G |
| 13B to 7B | Single Weights | Random | 27.06 | 39 min | 205 G |
| 13B to 7B | Single Weights | Sensitivity | 27.27 | 39 min | 205 G |
| 30B to 7B | Submatrix | Random | 28.11 | 82 min | 510 G |
| 30B to 7B | Submatrix | Sensitivity | 28.63 | 82 min | 510 G |
| 30B to 7B | Single Weights | Random | 27.12 | 82 min | 510 G |
| 30B to 7B | Single Weights | Sensitivity | 27.36 | 82 min | 510 G |

- For the structure of extracted parameters, "submatrix" extraction involves directly taking sub-matrices from the teacher model's larger matrix that aligns with the student model's dimensions. In contrast, "single weights" extraction means taking individual weights from the teacher model's larger matrix and arranging them into smaller matrices, maintaining their original order, to initialize the student model's LoRA module.

- We compare random selection with our sensitivity score-based method for choosing parameters.

- The "Score" column represents the average performance across four benchmarks.

- The running cost is compared on a CPU for 32 seed examples from GSM dataset, using teacher models of 13B and 30B. We perform backpropagation for each inference and based our experiments on fp32. Given that users may have limited GPU memory in real-world applications, we conduct these experiments on CPUs.

Key observations:

- The sensitivity score-based method we propose consistently outperforms random extraction.

- Extracting parameters via submatrices is significantly more effective than by single weights. This aligns with our discussion in Section 4.3 about the importance of maintaining the structural integrity of parameters for successful parametric knowledge transfer.

- Scaling up the teacher model from 13B to 30B consistently enhances performance but comes with about 2.1xthe runtime and 2.5x the memory usage.

- We can observe that the performance of transferring from 30B to 7B (Submatrix + Random) is comparable to (slightly lower than) 13B to 7B (Submatrix + Sensitivity). Thus, for applications with resource constraints, opting to randomly extract sub-matrices directly from a larger teacher model presents a viable alternative, considering it reduces inference time.

