# OpenReview forum: "Seeking Neural Nuggets: Knowledge Transfer in Large Language Models from a Parametric Perspective"
_ICLR.cc/2024/Conference — ICLR 2024 poster_

### Official Review · Reviewer_NWu9 · 2023-10-22

**Soundness:** 4 excellent
**Presentation:** 4 excellent
**Contribution:** 4 excellent
**Rating:** 8
**Confidence:** 1

**Summary:**

Large Language Models (LLMs) inherently encapsulate a vast reservoir of knowledge within their parameters, acquired through pre-training on extensive textual data. The authors investigate the possibility of transferring parametric knowledge across Large Language Models (LLMs) of different sizes. They introduce a novel approach, examining knowledge transfer from a unique parametric angle. Employing a two-stage framework involving knowledge extraction and injection, they conduct thorough experiments on four varied benchmarks, confirming the natural transferability of model parameters. Additionally, through detailed analysis of the pivotal factors impacting parametric knowledge transfer, they provide valuable insights for future research in this field.

**Strengths:**

The transferability of parametric knowledge is an important and timely topic that this paper addresses. Filling this gap is crucial in understanding the broader applicability and potential democratization of cutting-edge machine learning capabilities.

The paper introduces a novel parametric knowledge transfer paradigm. It moves beyond traditional methods like online and offline distillation and presents a unique approach focused on selecting specific static parameters from teacher models and injecting them into student models.

The experiments are conducted across various benchmark categories and LLM sizes. The results consistently demonstrate improvements in student performance after transferring teacher model parameters, providing strong empirical support for the proposed method.

The paper provides in-depth analysis of crucial factors such as teacher scales, initialization strategies, number of seed samples, and the nature of extracted parameters. This enriches the understanding of parametric knowledge transfer and its underlying dynamics.

The paper is clear and concise in writing.

**Weaknesses:**

How does your method compare with the existing distillation and pruning methods? It would have been great to see some other methods applied for comparison.

**Questions:**

Given the rapid evolution of LLMs, do you foresee the long-term viability and adaptability of the proposed knowledge transfer approach?

---

> ### Author Response · Authors · 2023-11-18
> **Responses to Reviewer NWu9 - Part 1**
>
> We are grateful for your valuable review and thoughtful suggestions, and we try to answer your questions as follows.
>
> > **Q1**: How does your method compare with the existing distillation and pruning methods? It would have been great to see some other methods applied for comparison.
>
> Firstly, we would like to clarify that our study's focus on parametric knowledge transfer is fundamentally distinct from distillation and pruning methods, characterized by the following differences:
>
> * **Purpose and Focus**: Rather than proposing better distillation or pruning methods, our study is focused on exploring the transferability of implicit knowledge embedded in static parameters. While prior research has explored the detectability and editability of parametric knowledge, its transferability remains less explored. Our experiments provide empirical evidence in the knowledge transfer scenario, where student models demonstrate improved performance after receiving task-specific knowledge from the teacher model, as shown in Tables 1 and 2.
>
> * **Process and Efficiency**: Our approach differs from standard knowledge distillation, which typically requires fine-tuning or the direct involvement of the teacher model in student model training — a computationally intensive process. In contrast, our parametric knowledge transfer involves extracting task-specific parameters from the vanilla teacher model and integrating them into the student model. This method, requiring only 32 inferences from the teacher model, offers a significant efficiency advantage, especially in the context of LLMs.
>
> Despite these differences, we recognize the relevance of distillation methods in the knowledge transfer scenario and have conducted comparative experiments using LLaMA-1 models (13B to 7B and 30B to 7B). Here KD refers to vanilla knowledge distillation, while SeqKD is sequence-level knowledge distillation [1].
>
> | Models          | GSM    | MMLU   | Super NI | AlpacaFarm | Average |
> |-----------------|--------|--------|----------|------------|---------|
> | Vanilla 7B      | 4.70   | 32.10  | 5.55     | -          | -       |
> | Vanilla 13B     | 4.93   | 43.50  | 7.78     | -          | -       |
> | 7B-LoRA         | 17.26  | 43.43  | 40.49    | 9.07       | 27.56   |
> | 13B to 7B (KD)   | 17.69  | 43.57  | 42.08    | 9.32       | 28.17   |
> | 13B to 7B (SeqKD)| 17.86  | 43.33  | 41.91    | 9.36       | 28.12   |
> | 13B to 7B (Ours) | **18.73**  | 44.03  | 42.37    | 9.28       | 28.60   |
> | 30B to 7B (KD)   | 17.81  | 44.10  | 41.96    | 9.48       | 28.34   |
> | 30B to 7B (SeqKD)| 17.99  | 43.97  | 42.40    | **9.61**       | 28.49   |
> | 30B to 7B (Ours) | 18.63  | **45.20**  | **43.08**    | 9.40       | **29.08**  |
>
> * All models are fine-tuned with LoRA, using identical training data and hyperparameters.
> * In this paper, we attempt to explore the evidence that knowledge in static parameters can be transferred between different LLMs, and knowledge transfer is the scenario in which we find and provide empirical evidence. Our focus is not on proposing to find better methods in this scenario.
>
> **References**:
>
> [1] Kim et al.  Sequence-level knowledge distillation. EMNLP 2016.

---

> ### Author Response · Authors · 2023-11-18
> **Responses to Reviewer NWu9 - Part 2**
>
> > **Q2**: Given the rapid evolution of LLMs, do you foresee the long-term viability and adaptability of the proposed knowledge transfer approach?
>
> As the landscape of LLMs continues to rapidly evolve, impacting model structures, scales, and the encoding of implicit knowledge, we believe the long-term viability and adaptability of our proposed knowledge transfer approach for several key aspects:
>
> * **Efficiency and Practicality**: The process of transferring knowledge using sub-matrix extraction and the LoRA-based approach is efficient, requiring only a small number of inferences from the teacher model (32 in our case). As LLMs grow in size and complexity, this kind of efficiency becomes ever more crucial.
>
> * **Scalability with Model Sizes**: Our method has demonstrated effectiveness in transferring knowledge between models of varying sizes, as shown in our experiments with different scales of LLaMA models (7B, 13B, 30B, and the newly added 70B). Moreover, as the teacher model's scale increases, our approach shows improved performance in knowledge transfer, aligning well with the ongoing trend of scaling up LLMs.
>
> * **Architecture Flexibility**: Our approach, grounded in the concept of parametric knowledge transfer, is not tightly bound to any specific model architecture. The underlying principles are adaptable to various forms of LLMs. As we demonstrated in our experiments with LLaMA models, our approach can handle different scales, even with different numbers of layers and dimensions or architectural differences (newly added LLaMA-2 70B to 7B). This flexibility suggests the method's potential adaptability to future LLM architectures.
>
> * **Methodological Flexibility**: Centered around the idea of intrinsic dimension and utilizing techniques like LoRA, our approach can be adjusted or expanded to incorporate other methods that refine matrix intrinsic dimensions. This adaptability makes our framework suitable for integration with future developments in LLM methodologies.
>
> * **Potential for Further Research**: Our research presents empirical evidence of the transferability of implicit knowledge within model parameters across various LLMs. This opens up avenues for further research, especially as LLMs increase in size, making fine-tuning or updating model parameters more challenging. Exploring how to better utilize knowledge within static parameters may become increasingly practical and significant.
>
> In conclusion, despite the rapid evolution of LLMs presenting new challenges, the core principles of our parametric knowledge transfer approach have the potential to remain applicable and adaptable in the long run.

---

> ### Author Response · Authors · 2023-11-22
> **Kind Reminder**
>
> Dear Reviewer NWu9,
>
> Thank you again for your valuable feedback and comments!
>
> As the discussion period is ending soon, we would greatly appreciate it if you could let us know whether you are satisfied with our response. We will be happy to address any remaining concerns.
>
> Sincerely,
>
> Paper 8932 Authors

---

> > ### Comment · Reviewer_NWu9 · 2023-11-22
> > **Thank you for your response**
> >
> > Dear Authors,
> >
> > Thank you for your response and responding to the question.
> >
> > Good luck !!

---

### Official Review · Reviewer_ccZi · 2023-10-23

**Soundness:** 4 excellent
**Presentation:** 3 good
**Contribution:** 3 good
**Rating:** 6
**Confidence:** 3

**Summary:**

The authors propose to better understand knowledge transfer between LLMs by identifying/extracting/aligning knowledge-specific parameters between teacher/upstream and student/downstream LLMs.
Compared to offlne/online distillation methods which transfer knowledge via synthetic datasets / teacher model outputs (offline) or use teacher activations as a loss/regularizer for the student (online), the proposed parametric knowledge transfer method seeks to identify and re-scale task-specific parameters in the teacher model and inject them into the student model via LoRA.
The authors use the gradient of parameters with respect to the loss across a number of sampled instances as a parameter set sensitivity measure, and extract parameters via 1) identifying the top LAYERS by task sensitivity; 2) extracting student matrix-size submatrices from the top teacher layer matrices to transfer; and 3) decompose the extracted parameters and fine-tune the student model with LoRA initialized from the decomposition.
The authors demonstrate that their method out-performs vanilla LoRA and fine-tuning of smaller student models across a variety of reasoning tasks.

**Strengths:**

S1 - The paper targets an important fundamental question about how knowledge is stored/represented across language models (e.g. are representations aligned / similarly structured across different pre-training or architectural regimes).

S2 - The authors evaluate on a good variety of knowledge-rich tasks (reasoning, MMLU/professional QA, instruction tasks, and open dialog) to demonstrate significant improvements over vanilla LoRA and fine-tuning. It would be interesting to see a wider range of model sizes being compared or comparing models pre-trained on different corpora (e.g. LLaMA-1/2 cross-transfer, or something like GPT-J).

S3 - There is a solid discussion of different factors that may affect parametric knowledge transfer here (layer selection, parameter location [ffn/self-attention], structure in transferred parameters, and number of seed samples)

**Weaknesses:**

W1 - The technical details (Section 3) are succintly written but could be written in a more clear and understandable way (e.g. summarize how standard LoRA is modified in knowledge injection). Certain details are explained in a more accessible manner in e.g. 4.3 but should be present in the main technical spec.

W2 - The diagrams can be more clearly framed with larger font and clear colors. In particular, the legend for Figure 3 is very difficult to read.

W3 - In 4.3, it seems that random submatrix initialization from the 30B model is copmarable to sensitivity-extracted submatrices from the 13B model. This merits further discussion about what the trade-offs are between just using random sub-matrices from larger models vs. the sensitivity method.

**Questions:**

Q1 - Can the authors provide more detail about 3.2 - how exactly the submatrix (with highest cumulative sensitivity) is extracted from the larger-parameter teacher model layer to create a mapping matrix in the student model's parameter set? And why in particular extracting a submatrix (with arbitrary possible location in the larger 2D teacher layer matrix) works?

---

> ### Author Response · Authors · 2023-11-18
> **Responses to Reviewer ccZi**
>
> Thanks for the detailed review and helpful comments. In response to your questions, our answers are as follows.
>
> > **Q1 & Q2**: The technical details could be written in a more clear and understandable way & the legend for Figure 3 is very difficult to read.
>
> Thank you for your valuable feedback, especially regarding the clarity of our technical writing and the legend of Figure 3!
>
> In response to your suggestions, we have revised the sections on sub-matrix extraction (Section 3.2) and knowledge injection using LoRA (Section 3.3) within the methodology part of our paper. Additionally, we have redesigned Figure 3 and updated its corresponding explanation in Section 4.3 of the analysis section. These modifications are highlighted in blue in our revised manuscript for your convenience.
>
> Please let us know if there are still parts that you find challenging to comprehend. We are committed to enhancing our presentation to ensure it is accessible and clear to all readers.
>
> > **Q3**: In 4.3, it seems that random submatrix initialization from the 30B model is comparable to sensitivity-extracted submatrices from the 13B model. This merits further discussion about what the trade-offs are between just using random sub-matrices from larger models vs. the sensitivity method.
>
> Firstly, we would like to clarify random extraction methods depicted in Figures 3 and 5(c) differ: Figure 3 illustrates random submatrix extraction, whereas Figure 5(c) demonstrates random single-weight extraction. Our findings indicate that extracting submatrices yields significantly better results, underscoring the importance of preserving the structural integrity of the teacher model's parameters in effective parametric knowledge transfer.
>
> Addressing your question, the comparable performance between the 30B to 13B (Random) and the 13B to 7B (Sensitivity) is indeed an intriguing point. However, it's essential to consider the overhead perspective. Our method requires only 32 inferences from the teacher model, meaning scaling up from 13B to 30B does not impose a significant additional computational burden. Moreover, this scaling up consistently enhances performance in three out of the four tasks we studied, as shown in Table 1. Therefore, while using randomly extracted submatrices from larger models can achieve satisfactory performance within our framework, we still recommend employing the seed samples-based method for parameter extraction that we propose for better results.
>
> > **Q4**: Can the authors provide more detail about 3.2 - how exactly the submatrix (with highest cumulative sensitivity) is extracted from the larger-parameter teacher model layer to create a mapping matrix in the student model's parameter set? And why in particular extracting a submatrix (with arbitrary possible location in the larger 2D teacher layer matrix) works?
>
> For the implementation of our method, we don't employ a mapping matrix. Instead, our process is as follows:
>
> 1. We go through all sub-matrices in the teacher model's parameter matrix that align with the dimensions of the student model. During this, we identify and extract the sub-matrix with the highest total sensitivity scores.
> 2. This chosen submatrix is then directly decomposed and used to initialize the LoRA module for the student model.
>
> We apologize for any confusion and have revised the descriptions in Sections 3.2 and 3.2 in the updated manuscript to clarify this process.
>
> As for why extracting sub-matrices proves more effective than selecting individual weights or rows and columns, our approach is inspired by research in structured pruning [1-2]. Studies in this area have shown that structurally pruning layers, heads, intermediate dimensions, or blocks in weight matrices is more effective than point-wise pruning. These findings suggest that task-specific knowledge is often more concentrated in complete structures within a network, rather than being dispersed across isolated weights. In our research on parametric knowledge transfer, we've observed similar patterns, affirming the efficacy of extracting and utilizing these integral sub-structures.
>
> **References**:
>
> [1] Lagunas et al. Block Pruning For Faster Transformers. EMNLP 2021.
>
> [2] Xia et al. Structured Pruning Learns Compact and Accurate Models. ACL 2022.

---

> ### Author Response · Authors · 2023-11-22
> **Kind Reminder**
>
> Dear Reviewer ccZi,
>
> Thank you again for your valuable feedback and comments!
>
> As the discussion period is ending soon, we would greatly appreciate it if you could let us know whether you are satisfied with our response. We will be happy to address any remaining concerns.
>
> Sincerely,
>
> Paper 8932 Authors

---

> > ### Comment · Reviewer_ccZi · 2023-11-22
> >
> > Thank you for the response to my questions - it's helpful in clarifying some of the points around submatrix extraction/initialization. I'm still a bit confused by W3 however and feel like it should be explained in more detail in the paper with accompanying discussions around what ramifications this has in real-world applications. As such, I have not revised my score.

---

> > > ### Author Response · Authors · 2023-11-22
> > > **Response to Reviewer ccZi**
> > >
> > > Thank you for your response. Based on your suggestion, we discuss the trade-offs between performance and running costs as follows:
> > >
> > > | Transfer | Structure of Ext. Para. | Methods for Ext. Para. | Score | Inference Time | CPU Memory |
> > > |----------|-------------------------|------------------------|-------|----------------|------------|
> > > | 13B to 7B | Submatrix | Random | 27.83 | 39 min | 205 G |
> > > | 13B to 7B | Submatrix | Sensitivity | 28.22 | 39 min | 205 G |
> > > | 13B to 7B | Single Weights | Random | 27.06 | 39 min | 205 G |
> > > | 13B to 7B | Single Weights | Sensitivity | 27.27 | 39 min | 205 G |
> > > | 30B to 7B | Submatrix | Random | 28.11 | 82 min | 510 G |
> > > | 30B to 7B | Submatrix | Sensitivity | 28.63 | 82 min | 510 G |
> > > | 30B to 7B | Single Weights | Random | 27.12 | 82 min | 510 G |
> > > | 30B to 7B | Single Weights | Sensitivity | 27.36 | 82 min | 510 G |
> > >
> > > **Experimental details**:
> > > * We conduct comparisons for two knowledge transfers: LLaMA-1 13B to 7B and 30B to 7B.
> > > * For the structure of extracted parameters, "submatrix" extraction involves directly taking sub-matrices from the teacher model's larger matrix that aligns with the student model's dimensions. In contrast, "single weights" extraction means taking individual weights from the teacher model's larger matrix and arranging them into smaller matrices, maintaining their original order, to initialize the student model's LoRA module.
> > > * We compare random selection with our sensitivity score-based method for choosing parameters.
> > > * The "Score" column represents the average performance across four benchmarks.
> > > * The running cost is compared on a CPU for 32 seed examples from the GSM dataset, using teacher models of 13B and 30B. We perform backpropagation for each inference and based our experiments on fp32. Given that users may have limited GPU memory in real-world applications, we conduct these experiments on CPUs.
> > >
> > > **Key Observations**:
> > > * The sensitivity score-based method we propose consistently outperforms random extraction.
> > > * Extracting parameters via submatrices is significantly more effective than by single weights. This aligns with our discussion in Section 4.3 about the importance of maintaining the structural integrity of parameters for successful parametric knowledge transfer.
> > > * Scaling up the teacher model from 13B to 30B consistently enhances performance but comes with about 2.1xthe runtime and 2.5x the memory usage.
> > > * As mentioned by the reviewer, the performance of transferring from 30B to 7B (Submatrix + Random) is comparable to (slightly lower than) 13B to 7B (Submatrix + Sensitivity). Thus, for applications with resource constraints, opting to randomly extract sub-matrices directly from a larger teacher model presents a viable alternative, considering it reduces inference time.
> > >
> > > Thank you once more for your insightful feedback and valuable suggestions, which have greatly contributed to enhancing our paper. We concur with your recommendation that discussing the trade-offs between running costs and performance is both useful and essential, especially in the context of application scenarios with varying resource constraints. To address this, we have incorporated this discussion into Appendix A.5 of our revised paper. If there are any additional questions or aspects you feel need further clarification, please feel free to let us know.

---

> > > > ### Comment · Reviewer_ccZi · 2023-12-05
> > > >
> > > > Thank you for the follow-up answer! This addresses some of my concerns and I have raised my Soundness score to 4.

---

### Official Review · Reviewer_Df53 · 2023-11-07

**Soundness:** 3 good
**Presentation:** 3 good
**Contribution:** 3 good
**Rating:** 6
**Confidence:** 4

**Summary:**

This paper proposes a method to transfer knowledge from big to small models by sharing the parameter. To be specific, it first detects and extracts a submatrix of parameters in large models, and the submatrix has the same size as the small model. Then, it decomposes the matrix into the Lora module and adds this module with the parameters of a small model. On Five datasets, this method brings about 2 points improvement compared to the original small model.

**Strengths:**

1. This paper proposes a novel method that transfers knowledge from large models to small models by sharing parameters.

**Weaknesses:**

1. The experiments are limited. This paper only tests on a few datasets for LLMs, does not discuss which amount of parameters is the best for transferring, and does not compare with other distillation methods, and does not compare with other methods about intrinsic dimension.

2. The application of this method is limited. The paper does not experiment on different models. Can the knowledge transfer across architectures with this method?  This paper also assumes that the small model is already fine pre-trained. Can we transfer to the random-initialized model?

**Questions:**

1. I would like to see results from llama2 70b to 7b.

2. In Formula 3, you use Taylor expansion to approximate the change of masking some parameters. But mask parameters are not trivial. Why can 1-order Taylor expansion approximate it?

---I have read the whole rebuttal and raise my score from 5 to 6.

---

> ### Author Response · Authors · 2023-11-18
> **Responses to Reviewer Df53 - Part 1**
>
> We appreciate your thorough review as well as constructive feedback, and we try to answer your questions as follows.
>
> > **Q1**: I would like to see results from llama2 70b to 7b.
>
> We add the following experiment based on your suggestion, where for the GSM and MMLU datasets we present the 0-shot results and the metric used for Super NI is ROUGE-L.
>
> | Models      | GSM    | MMLU   | Super NI | AlpacaFarm | Average |
> |-------------|--------|--------|----------|------------|---------|
> | Vanilla 7B  | 3.34   | 41.70  | 4.68     | -          | -       |
> | Vanilla 13B | 6.52   | 52.10  | 4.84     | -          | -       |
> | 7B-LoRA     | 23.38  | 47.77  | 41.25    | 20.50      | 33.23   |
> | &nbsp;&nbsp;&nbsp;&nbsp;+13B Param.  | 25.30  | 49.37  | 42.98    | **24.64**      | 35.57   |
> | &nbsp;&nbsp;&nbsp;&nbsp;+70B Param. | **26.16**  | **49.60**  | **43.65**    | 24.27      | **35.92**   |
>
> * Due to the differences in attention architectures between LLaMA-2 70B and 7B models (with the 70B employing grouped-query attention and the 7B using multi-head attention), we restrict the parametric knowledge transfer from 70B to 7B to the FFN and embedding layers, which account for 0.36% trainable parameters. In contrast, in the transfer from the 13B to 7B model, we also include the attention module, increasing the percentage of trainable parameters to 0.61%.
> * Nevertheless, the transfer from the 70B to the 7B model demonstrates greater performance gains than transferring from 13B to 7B. This implies that our parametric knowledge transfer approach becomes increasingly effective as the teacher model scales up, even in the presence of architectural differences beyond the number of layers and dimensions.
>
> > **Q2**: In Formula 3, you use Taylor expansion to approximate the change of masking some parameters. But mask parameters are not trivial. Why can 1-order Taylor expansion approximate it?
>
> The choice to neglect the first-order Taylor's remainder, which could be computed for greater accuracy using the Lagrange form, is based on practical considerations. Firstly, incorporating this remainder entails substantial computational efforts. Secondly, activation function such as ReLU typically leads to a smaller second-order term [1]. Therefore, this approximation method has been extensively employed in previous research and has consistently demonstrated effective results [2-4].
>
> In addition, to further verify the efficacy of our sensitivity-based approach, we have included visualizations in Appendix A.1 of the revised version (Figure 6). They illustrate how parametric knowledge is distributed across different layers in four distinct task categories, which helps showcase the ability of our approach to capture task-specific knowledge.
>
> > **Q3**: Discussion about which amount of parameters is the best for transferring.
>
> Our analysis of the origin of parameters (Section 4.3, Figure 5(b)) involves a discussion of parameter amounts. For example, transferring a combination of the embedding layer, FFN, and attention modules, which constitute 0.608% of the model's parameters, yields the best results. In contrast, transferring a single module, like the FFN alone which accounts for 0.343%  trainable parameters, leads to relatively poor performance.
>
> To further investigate the effect of the amount of parameters, we extend the experiments at LLaMA-1 13B to 7B and 30B to 7B by introducing comparisons with different LoRA r (intrinsic rank). The results are the average scores on the four datasets.
>
> | Transfer Module | LoRA r | Param. | 13B to 7B | 30B to 7B |
> |-----------------|--------|--------|----------|----------|
> | Embedding       | 128    | 0.137% | 27.33    | 27.52    |
> | Attention       | 16     | 0.248% | 27.53    | 27.92    |
> | FFN             | 16     | 0.343% | 27.87    | 28.23    |
> | All             | 4      | 0.152% | 27.93    | 28.37    |
> | All             | 8      | 0.304% | 28.17    | 28.48    |
> | All             | 16     | 0.608% | 28.22    | **28.63**    |
> | All             | 32     | 1.216% | 28.19    | 28.54    |
> | All             | 64     | 2.432% | **28.27**    | 28.60    |
>
> * For the transfer involving only the embedding layer, we set LoRA r at 128, ensuring that the number of trainable parameters remains comparable.
> * We observe that increasing LoRA r beyond 16 does not significantly enhance the results when transferring all modules. Consequently, we maintain LoRA r at 16 for the experiments presented in this paper.
> * Our analysis indicates that the origin of the parameters has a more pronounced impact on the effectiveness of parametric knowledge transfer than the amount of parameters transferred.

---

> ### Author Response · Authors · 2023-11-18
> **Responses to Reviewer Df53 - Part 2**
>
> > **Q4**: Comparison with distillation methods.
>
> To address your query, it's important to first emphasize that our study's focus on parametric knowledge transfer is fundamentally distinct from traditional distillation methods, characterized by the following differences:
>
> * **Purpose and Focus**: Rather than proposing better distillation methods, our study is focused on exploring the transferability of implicit knowledge embedded in static parameters. While prior research has explored the detectability and editability of parametric knowledge, its transferability remains less explored. Our experiments provide empirical evidence in the knowledge transfer scenario, where student models demonstrate improved performance after receiving task-specific knowledge from the teacher model, as shown in Tables 1 and 2.
>
> * **Process and Efficiency**: Our approach differs from standard knowledge distillation, which typically requires fine-tuning or the direct involvement of the teacher model in student model training — a computationally intensive process. In contrast, our parametric knowledge transfer involves extracting task-specific parameters from the vanilla teacher model and integrating them into the student model. This method, requiring only 32 inferences from the teacher model, offers a significant efficiency advantage, especially in the context of LLMs.
>
> Despite these differences, we recognize the relevance of distillation methods in the knowledge transfer scenario and have conducted comparative experiments using LLaMA-1 models (13B to 7B and 30B to 7B). Here KD refers to vanilla knowledge distillation, while SeqKD is sequence-level knowledge distillation [5].
>
> | Models          | GSM    | MMLU   | Super NI | AlpacaFarm | Average |
> |-----------------|--------|--------|----------|------------|---------|
> | Vanilla 7B      | 4.70   | 32.10  | 5.55     | -          | -       |
> | Vanilla 13B     | 4.93   | 43.50  | 7.78     | -          | -       |
> | 7B-LoRA         | 17.26  | 43.43  | 40.49    | 9.07       | 27.56   |
> | 13B to 7B (KD)   | 17.69  | 43.57  | 42.08    | 9.32       | 28.17   |
> | 13B to 7B (SeqKD)| 17.86  | 43.33  | 41.91    | 9.36       | 28.12   |
> | 13B to 7B (Ours) | **18.73**  | 44.03  | 42.37    | 9.28       | 28.60   |
> | 30B to 7B (KD)   | 17.81  | 44.10  | 41.96    | 9.48       | 28.34   |
> | 30B to 7B (SeqKD)| 17.99  | 43.97  | 42.40    | **9.61**       | 28.49   |
> | 30B to 7B (Ours) | 18.63  | **45.20**  | **43.08**    | 9.40       | **29.08**  |
>
> * All models are fine-tuned with LoRA, using identical training data and hyperparameters.
> * In this paper, we attempt to explore the evidence that knowledge in static parameters can be transferred between different LLMs, and knowledge transfer is the scenario in which we find and provide empirical evidence. Our focus is not on proposing to find better methods in this scenario.
>
> > **Q5**:  Discussion about other methods for intrinsic dimension.
>
> Regarding the intrinsic dimension aspect, our focus isn't to develop or compare various methods in this area. Instead, we propose using the concept of intrinsic dimension as a bridge for transferring parametric knowledge. We instantiate our framework with LoRA, a standard representative in this domain. The process involves decomposing the teacher model's parameter matrix, which then serves as the initialization for the student model's LoRA module, aiding in the knowledge injection.
>
> Notably, the proposed workflow is designed to be adaptable, allowing any method that refines the intrinsic dimension of matrices to be directly incorporated into our existing framework.
>
> > **Q6**: About “this paper only tests on a few datasets for LLMs”.
>
> We would like to highlight the diversity and comprehensiveness of the 4 datasets we selected. 3 of these are collections of multiple tasks or domains: MMLU includes questions from 57 varied domains, Super NI covers 1,616 different NLP tasks, and AlpacaFarm is a compilation of open-ended conversation sources from five different origins. Given their diversity and extensive use in evaluating LLMs, we believe that the consistent improvements demonstrated by our method across these varied categories of benchmarks are a strong indication of its effectiveness.

---

> ### Author Response · Authors · 2023-11-18
> **Responses to Reviewer Df53 - Part 3**
>
> > **Q7**: The paper does not experiment on different models. Can the knowledge transfer across architectures with this method?
>
> We answer this question from the following two perspectives:
>
> * **Previous Research on Parametric Knowledge**: Earlier studies have primarily focused on detecting and editing parametric knowledge within a single pre-trained language model. When it comes to merging models, the common practice has been to use two identical models. As a result, the potential for transferring parametric knowledge between models with different layers, dimensions, or overall architectures remains largely unexplored.
>
> * **Our Experimental Models**: In our experiments, we use LLaMA-1 and LLaMA-2 models, which differ in their pre-training. The LLaMA models come in various scales (7B, 13B, 30B, 70B), each with a different number of layers and dimensions. In addition, the LLaMA-2 70B model has a unique attention module. Despite these significant differences, our experiments successfully demonstrate the transferability of parametric knowledge across various models. Moreover, our supplementary experiments transferring knowledge from LLaMA-2 70B to 7B (see Q1) confirm that our method remains effective even when attention architectures differ. Given that most Transformer variants share the same architecture in terms of FFNs and embedding layers, our method appears viable for transferring parametric knowledge across these models.
>
> > **Q8**: This paper also assumes that the small model is already fine pre-trained. Can we transfer to the random-initialized model?
>
> Recent research on knowledge transfer in the context of PLMs and LLMs typically assumes that student models are fine pre-trained. This is primarily due to two factors:
>
> 1. From a computational perspective, the cost of pre-training from scratch is prohibitive.
> 2. For practical applications, the majority of task-specific models are built upon and fine-tuned using existing PLMs or LLMs, leading to a research focus on the fine-tuning stage.
>
> However, we strongly agree that the study on the random-initialized models has promising applications, and we discuss this direction of research in our related work (Section 3.2), termed knowledge inheritance. This concept involves larger models receiving parameters from smaller models to accelerate the pre-training process. This research direction contrasts with the focus of our paper, as it emphasizes a different phase of training (pre-training rather than fine-tuning) and a reverse direction of knowledge flow (from smaller to larger models, as opposed to our large-to-small transfer).
>
> Thank you once again for your valuable feedback and comments! They have been crucial in enhancing the comprehensiveness of our experiments and enriching our study with deeper insights and more empirical explorations in the field of parametric knowledge transfer. If there are any further questions or aspects you feel remain unaddressed, we are more than willing to provide additional information and clarifications as needed.
>
> **References**:
>
> [1] Molchanov et al. Pruning Convolutional Neural Networks for Resource Efficient Inference. ICLR 2017.
>
> [2] Ding et al. Global Sparse Momentum SGD for Pruning Very Deep Neural Networks. NeurIPS 2019.
>
> [3] Lubana et al. A Gradient Flow Framework For Analyzing Network Pruning. ICLR 2021.
>
> [4] Liang et al. No Parameters Left Behind: Sensitivity Guided Adaptive Learning Rate for Training Large Transformer Models. ICLR 2022.
>
> [5] Kim et al.  Sequence-level knowledge distillation. EMNLP 2016.

---

> ### Author Response · Authors · 2023-11-22
> **Kind Reminder**
>
> Dear Reviewer Df53,
>
> Thank you again for your valuable feedback and comments!
>
> As the discussion period is ending soon, we would greatly appreciate it if you could let us know whether you are satisfied with our response. We will be happy to address any remaining concerns.
>
> Sincerely,
>
> Paper 8932 Authors

---

> > ### Comment · Reviewer_Df53 · 2023-12-05
> > **Response to rebuttal**
> >
> > In the authors' response, this paper has included comparisons in 70b to 7b settings and comparisons with other distilling methods. However, it depends on the training data and iterations; I doubt that this method can still outperform distilling methods at a larger scale of data. I agree with the rebuttal to the questions regarding the number of datasets and the number of hyperparameters( I suggest you list the raw baseline).  I am not satisfied with the questions regarding the Taylor expansion, intrinsic dimension, experiment on different models, and transferring to random models. Therefore, the rating of this is borderline. Given the author's response and other reviews, I decided to raise the rating from borderline reject to borderline accept.

---

### Author Response · Authors · 2023-11-18
**General Response**

Dear Reviewers,

We are deeply grateful for your insightful feedback and valuable suggestions. Based on your reviews, we have made thorough revisions to our manuscript, highlighting these changes in blue for clarity.

We also wish to express our appreciation for your recognition of the strengths of our work, including:

* Our paper targets an important, fundamental, and timely topic (ccZi, NWu9)
* Our proposed paradigm is novel (Df53, NWu9)
* Evaluating on a good variety of knowledge-rich tasks (ccZi, NWu9)
* Our approach consistently demonstrates improvements (Df53, ccZi, NWu9)
* In-depth analysis of the crucial factors for parametric knowledge transfer (ccZi, NWu9)
* The paper is clear and concise in writing (NWu9)

We have provided specific responses to each reviewer’s questions separately. In the revised submission, we have integrated your suggestions, which include additional clarifications of our method and figure, expanded experimental results, and new visualizations. Below is a summary of the key updates:

* Added visualization for parametric knowledge [Appendix A.1, Figure 6]
* Expanded experiments for LLaMA-2 70B to 7B [Appendix A.2, Table 3]
* Included experiments on the amount of parameters [Appendix A.3, Table 4]
* Added discussion and comparison with distillation methods [Appendix A.4, Table 5]
* Improved and polished the methodology section [Sections 3.2 and 3.3]
* Revised figures for better clarity and understanding [Section 4.3 and Figure 3]
* Trade-off discussion between performance and running cost [Appendix A.5]

We sincerely thank you again for your contributions to improving our work. If there are any further concerns or queries, we are fully prepared to address them.

---

### Meta-Review · Area_Chair_RNYp · 2023-12-12

**Metareview:**

This paper develops a novel parametric knowledge transfer method in LLMs, which distills knowledge from a larger model towards a smaller one. The main idea is to extract knowledge-pertinent model parameter matrices via perturbation analysis and then inject into LoRA update via matrix factorization. All reviewers agree that this is an interesting and novel contribution to the problem. The authors have sufficiently discussed and addressed the concerns raised by the reviewers by adding additional experiments. The weakness of the initial version of the paper is lack of sufficient comparison with other distillation method. Although the method is proposing a new parametric transfer method, which is quite a novelty attempt to he problem, it is still important to compare to existing distillation method to see how it actually performs relative to other paradigms. This issue has been partially addressed during the rebuttal period by adding results of other methods, although the performance gain is relatively small. Another issue is since the advantage of parametric knowledge transfer is to transfer specific domain knowledge in the original teacher model, the paper has less discussion and experimental study on this side. Nevertheless, given the work opens a brand new avenue in this direction, it is worth publication in the conference.

**Justification For Why Not Higher Score:**

This paper has certain points that should be further addressed -- see the above meta review -- in order to be bumped up into a higher tier (spotlight or oral).

**Justification For Why Not Lower Score:**

The work opens a brand new avenue in this direction, and it is worth publication in the conference

---

### Decision · Program_Chairs · 2024-01-16

Accept (poster)